# Collective Responsibility in the Cooperative Governance of Climate Change

**Alessandro Piazza** 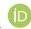

META Social Sciences and Humanities for Science and Technology Unit, Polytechnic University of Milan, 20133 Milan, Italy; alessandro.piazza@polimi.it

**Abstract:** This paper sets out a proposal for framing collective responsibility as a central element within the cooperative governance of climate change. It begins by reconstructing the analysis of climate change as a Tragedy of the Commons in the economic literature and as a Problem of Many Hands in the ethical literature. Both formalizations are shown to represent dilemmatic situations where an individual has no rational incentive to prevent the climate crisis and no moral requirement to be held responsible for contributing to it. Traditionally both dilemmas have been thought to be solvable only through a vertical structure of decision-making. Where contemporary research in political economy has undergone a "governance revolution", showing how horizontal networks of public, private, and civil society actors can play an important role in the management of the climate crisis, little research has been carried out in the ethical field on how to secure accountability and responsibility within such a cooperative structure of social agency. Therefore, this paper contributes by individuating some conditions for designing responsible and accountable governance processes in the management of climate change. It concludes by claiming that climate change is addressable only insofar as we transition from a morality based on individual responsibility to a new conception of morality based on our co-responsibility for preventing the climate crisis.

**Keywords:** cooperative governance; governance networks; social ontology; shared agency; collective responsibility; problem of many hands; tragedy of the commons

## 1. Introduction

To a large degree, the mitigation of the effects of climate change represents the greatest ethical and political challenge that our society faces today. The urgency of taking tempestive and effective climate action has been recognized by the United Nations as one of the key goals for sustainable development [1]. As the Intergovernmental Panel on Climate Change (IPCC) has claimed, "each of the last three decades has been successively warmer at the Earth's surface than any preceding decade since 1850" and according to the most up to date climate data analyzed by the World Meteorological Organization, the "average temperatures for the five-year, 2015-to-2019, and 10-year, 2010-to-2019, periods are almost certain to be the highest on record" [2]. Anthropogenic emissions of greenhouse gases are the main drivers of such an increase in global temperatures and they derive from increased energy consumption, industrial development, growing demographic numbers, land-use change, and consumption habits. To maintain the commitments of the Paris Agreement of limiting the increase in global average temperatures to 1.5 °C with respect to preindustrial levels, governments have to accelerate the transition toward sustainable development. However, the management of such transition pathways to "deep decarbonization" requires the coordination of complex socio–technical–ecological systems, which are characterized by the intertwinement of natural ecosystems, institutional regulations, private markets, infrastructures, technological innovations, and user practices [3,4]. As Oran Young has recognized, "sustainable development is a broad objective that calls for a melding of economic, social, and environmental factors, both to enhance the well-being of individual humans

and to produce resilient socio-ecological systems from the local to the global level" [5]. The management of such complex adaptive systems [6], which involves the expertise necessary for organizing the layered composition of technical, economic, environmental, and social challenges, is no longer within the reach of central administrations within nation states. To a large extent, traditional command-and-control practices are proving to be only a partial solution to the challenge of governing the complexity of the sustainable transition [7]. Within the academic literature, a variety of new approaches for the management of social–ecological systems has emerged: from *polycentric governance*, which is centered around the multiple and nested centers of decision-making involved in devising context-specific solutions to environmental problems [8,9], to *adaptive governance*, which is based on the dynamic capacity of social networks to self-organize, share knowledge, and respond adaptively to emergent social–ecological phenomena [10,11], to *collaborative governance*, which is grounded in the ability of multiple stakeholders, both public and private, to effectively share information and mutually learn from best practices in the achievement of common societal goals [12,13]. All of these approaches have emerged as an answer to the shortcomings of centralized regulation and downstream implementation in managing social–ecological systems, and they have contributed to a shift in the academic discourse toward cooperative and participatory models of governance. The advantages of these governance networks are the increased ability to adapt quickly to emergent phenomena, to provide fine-grained information on local impacts, to deploy articulated expertise in technological innovations, and to allow for effective multi-level coordination across government scales. In fact, as the scale and complexity of policy problems has increased exponentially, public policy has undertaken a "governance revolution" [14], where a vertical and centralized conception of public administration, focused on the *structure* of government, has been gradually supplanted by a horizontal and decentralized model of governance, centered instead on the *process* of governing, opening the management of policy problems to governance networks of societal actors from public, private, and civil society sectors [15]. This shift to the cooperative management of social–ecological systems has nonetheless brought about new challenges: a less structured decision-making process, a multiplicity of actors with diverging perspectives and interests, and the necessity of a continuous reciprocal adaptation of plans and policies. Therefore, the moral question of a sustainable future is centered around the successful management of the increasingly complex nature of the Earth system's governance [16]. The responsibility toward present and future generations for a sustainable transition forces all societal actors to address the question of how to achieve responsible collective agency. Hence, this article will concentrate on how governance can today answer to the planetary crisis that is climate change; at its center, this paper outlines two main challenges that a theory of governance has to meet when managing the effects of anthropogenic global warming: the fragmentation of agency between a collection of self-interested societal actors [17] and the resulting risk of failing to achieve any meaningful form of responsibility. A promising solution lies in creating a full theory of responsible cooperative governance within the management of social–ecological systems.

## 2. Methodology

This paper develops by modeling climate change as an instance of the Tragedy of the Commons in the economic literature [18] and as a Problem of Many Hands in the ethical literature [19,20]. Within the economic literature, much work has been carried out on the formalization of climate change as a commons dilemma [21–24]. Here, I will first offer a reconstruction of Garrett Hardin's original argument, and I will proceed to adopt Elinor Ostrom's account of commons dilemmas, which in many ways reformulates the initial set of assumptions present in Garrett Hardin's original work. In particular, I will show that Ostrom's theoretical and empirical work has contributed to questioning two main assumptions, framed within rational choice theory, which inform Hardin's reading of commons dilemmas: the absence of communication between players and the exclusively self-interested and utilitarian character of individual rationality. For what concerns the

formalization of climate change as a Problem of Many Hands, I will follow the work of van de Poel in his *The Problem of Many Hands: Climate Change as an Example* [25].

As the paper aimed at establishing a parallel between the rational dilemma that is the Tragedy of the Commons and the moral dilemma that is the Problem of Many Hands, its structure will alternate, in an ABAB scheme, between paragraphs devoted to the economic analysis of climate change and paragraphs devoted to its ethical discussion.

## 3. Materials

### 3.1. An Economic Formalization of Climate Change: The Tragedy of the Commons

It is first important to sketch in further detail how climate change has been formalized, inside economic theory, as a problem of "common resources" management in Garret Hardin's 1968 article *The Tragedy of the Commons* [18]. That paper, framed within a Malthusian logic [26], addresses one main challenge for our civilization: Earth is becoming too densely populated, which puts an unprecedented burden on our shared resources, namely, the commons. The core of Hardin's argument is to be found in the theoretical impasse reached in managing common-pool resources within a model of individual rationality. The argument develops by drawing a now renowned scenario: a group of herders lets their herds graze in a common pasture. Each herder will try to rationally maximize his utility by steadily growing his herd; at a certain moment, though, a certain threshold will be reached and an additional increment of one animal to the field will incur in the overgrazing of the pasture. At this point, so Hardin's argument goes, the addition of one animal will represent, for each herder, both a positive and a negative component of utility [18]:

> *The positive component is a function of the increment of one animal. Since the herdsman receives all the proceeds from the sale of one additional animal, the positive utility is nearly +1.*

> *The negative component is a function of the additional overgrazing created by one more animal. Since, however, the effects of overgrazing are shared by all the herdsmen, the negative utility for any particular decision-making herdsman is only a fraction of −1.*

As the scenario shows, the depletion of a common-pool resource occurs when the resource stock is consumed by the appropriators faster than its regeneration rate [27]. Nonetheless, irrespective of the consequences, in economic terms for each herdsman, the marginal benefits of adding cattle to the graze are larger than the marginal costs. Therefore, the rational conclusion to be drawn by any herdsman, faced with a decision between cooperating or defecting in the collective action, will be to "free-ride" and unilaterally choose what is in his best interest; the result will be the gradual addition of cattle to the pasture, with the further consequence of eventually depleting the commons. For Hardin, this shows how a model of individual rationality, applied to the management of a common resource, results in its eventual depletion: this fact constitutes the conceptual core of the Tragedy of the Commons. As Hardin commented, in commons dilemmas, we face a tragic situation where we lack a solution that has a "technical" character [18]:

> *A technical solution may be defined as one that requires a change only in the techniques of the natural sciences, demanding little or nothing in the way of change in human values or ideas of morality.*

It is possible to describe this technical failure in a game-theoretic language by framing the common pasture as an interactive decision-making game, where the optimal choice at the individual level paradoxically constitutes a suboptimal choice at the collective level [27]. As a result, in a Tragedy of the Commons, each herdsman, who is in the dark with respect to the other herdsmen's decisions, has an incentive to unilaterally defect, or "free-ride", rather than cooperating; this failure of coordination in a collective action ultimately results in an outcome that is not an equilibrium, and thus represents a cooperation problem [28]. As Hardin saw, this failure to cooperate would eventually place unsustainable pressure on common resources. Nowadays, an infinite number of tragedies of the commons, caused by unilateral and self-interested decision-making, feeds

the daily reports on the ongoing catastrophe that is climate change: oceans are undergoing progressive acidification, human and non-human life is threatened by the erosion of natural habitats, the atmosphere is becoming increasingly polluted, and global temperatures are rising. At every level, from nation states to city administrations, private companies, and consumer habits, human conduct is proving to be dramatically inadequate to prevent the depletion of our commons, bringing about an environmental and social disaster of unprecedented dimension. What Hardin provides is a game-theoretic analysis of such a disaster, showing how an insular model of *homo economicus*, moved by the maximization of individual utility, is bound to meet his anthropological limits when faced with the problem of managing a common resource.

### 3.2. An Ethical Formalization of Climate Change: The Problem of Many Hands

Coming to the ethical analysis of the Tragedy of the Commons, this section tackles the problem of climate change in terms of our moral responsibility to prevent it. The aim will be to argue how commons dilemmas constitute not only a rational impasse but also a moral one, as climate change can be modeled as a case of "collective responsibility without individual responsibility". This responsibility gap constitutes what, in the literature, has been called the Problem of Many Hands. Here, the meaning of responsibility will be taken as close to its etymological sense, as answering for one's actions; specifically, agents will be regarded as responsible for an action φ, when they are causally linked with a harm that they cannot reasonably justify, making their action blameworthy in an objective-reasons implying sense [29,30]. Where responsibility is usually framed within the context of past actions, in terms of remedial responsibility, here responsibility will be analyzed not in its backward-looking sense, but rather its forward-looking sense, as a form of prospective responsibility: we bear such a responsibility when we should prevent some event to bring about a bad outcome. To the degree that climate change poses an unprecedented threat to both present and future life on Earth, it can be maintained that preventing its devastating impacts represents a clear case of prospective responsibility [31]. Within metaethical theory, Ibo van de Poel has suggested that we hold a prospective responsibility (PR) when the following conditions apply [20,32]:

1. *Capacity condition*: the agent is capable of moral agency;
2. *Causal efficacy condition*: the agent is causally efficacious in producing the outcome;
3. *Normative condition*: bringing about the outcome is morally wrong.

Let us now examine how these three conditions apply to the actions of individuals in the case of climate change as an instance of a Tragedy of the Commons. Starting from the *capacity condition*, the attribution of moral capacity is regarded as a fundamental attribute of every person capable of intentional action. To the degree that an agent is capable of intentional agency, it can be claimed that such a person satisfies the capacity condition. As for the *causal efficacy condition*, it is possible to ask: can individuals prevent the depletion of the shared resource in a Tragedy of the Commons? Baylor Johnson, in *Ethical Obligations in a Tragedy of the Commons*, has convincingly argued that it seems difficult [33]. Looking closely at Johnson's argument, it is possible to see that the main point centers around the impossibility of being causally efficacious in non-coordinated agency:

> [ . . . ] *voluntary, unilateral reductions of use have no reasonable expectation of success when the situation faced strongly resembles a Tragedy of the Commons in other respects. It is very unlikely that most commons users will adopt such widespread restraint without organized assurances that others will mirror one's own restraint. The reasons are those given above: the incentives users have in such cases; each user's knowledge that her restraint is likely only to reward less scrupulous users; each user's awareness that every other user sees the same discouraging prospect; the need for nearly universal restraint in order to effectively protect the commons or reassure users that their sacrifice is not in vain.*

As it appears clear from the excerpt, what determines the absence of causal efficacy is not just the limited agency of the person but also the structure of the coordination game that every actor faces. Indeed, many philosophers have held that no individual person can be reasonably regarded as causally efficacious in preventing climate change [20,34–36]. Coming to the third, and final, *normative condition* for prospective responsibility, it can be asked whether any individual actor is engaging in some form of wrong-doing. Within the field of climate ethics, Walter Sinnott-Armstrong has argued, in his *It's Not My Fault: Global Warming and Individual Obligations*, that no individual actor can be held responsible for a form of wrong-doing in the case of climate change [34]. Here, the author claims, no plausible moral principle can determine a wrong-doing in failing to limit our carbon footprint, since individuals are neither sufficient nor necessary for determining global warming as a harm, individuals act under no intention of harming, and individual harms cannot be simply aggregated since global warming is an emergent, threshold phenomenon. A similar point is made by Johnson by arguing how an individual does not engage in wrong-doing in a commons dilemma because unilateral restrictions cannot be effective in preventing the depletion of the resource, the moral duty to unilaterally restrict the consumption of the resource might be overridden by the sacrifice and competitive disadvantage it entails, and finally, no one person's use of the commons is large enough to cause its depletion [33]. Therefore, it seems that from a moral perspective, no forward-looking responsibility can be attributed to individuals for preventing the depletion of our planetary resources. I submit that this fact constitutes a form of Moral Tragedy of the Commons. Conversely, from the point of view of the collective, these three conditions seem to be met: regarding the *capacity condition*, as long as humanity achieves some form of coordinated agency, it can be regarded as capable of intentional and moral agency; regarding the *causal efficacy condition*, as a collective, humanity can be causally efficacious in preventing climate change; finally, regarding the *normative condition*, as a whole, humanity can be considered morally blameworthy for bringing about the devastating intergenerational crisis that is climate change. As it appears from the reconstruction proposed, it can be advanced that there is symmetry in a Tragedy of the Commons between the dilemmatic disconnect between individual and collective rationality in its economic formalization and between individual and collective responsibility in its ethical formalization: just as there are collective reasons, but not individual ones, to prevent the depletion of common resources, there are collective moral reasons, but not individual ones, for preventing the disastrous effects that climate change will bring about. This dilemmatic situation, in which we have a fundamental gap between individual and collective responsibility, was first introduced by Dennis Thompson in *Moral Responsibility and Public Officials* as the Problem of Many Hands [19]. According to Ibo van de Poel, the Problem of Many Hands can be defined as follows [20]:

> The Problem of Many Hands (PMH) occurs if a collective is morally responsible for φ whereas none of the individuals making up the collective is morally responsible for φ.

Therefore, it can be argued that the Problem of Many Hands provides a useful ethical formalization of commons dilemmas, as in the case of climate change. As it appears from the reconstruction that has been proposed, we can advance the thesis that whenever a rational failure, such as a Tragedy of the Commons occurs, a parallel moral failure occurs as a Problem of Many Hands, since "free-riding" the commons is not irrational or irresponsible at the individual level, while it constitutes a rational and moral failure at the collective level.

### 3.3. Conventional Solutions to the Tragedy of the Commons: Governments and Markets

Within the field of economics, to face the structural shortcomings of collective action in commons dilemmas, two proposals have traditionally been advanced, both of which are grounded on the establishment of institutions: on the one hand, the appeal to the institution of the state, by turning the commons into a public good; on the other hand, the appeal to the institution of private property, by turning the commons into a private good. Where, in the first case, the structure of the coordination game gets changed through

the power of the state by introducing sanctions that modify the structure of individual incentives for defecting in the mutual effort and deviating from the equilibrium; in the second case, the coordination problem is solved by eliminating the very necessity of a collective action, as the commons get partitioned between the different actors, and the role of coordination is thereafter provided by the market. As Elinor Ostrom pointed out, the debate revolved for the better part of the 1970s and 1980s around a fundamental opposition between defenders of the "market" formula and supporters of the "Leviathan" solution [21]. On both views, the failure of individual rationality in a commons dilemma requires the creation of an external institution to enforce rules on the actors to prevent their eventual depletion. Hardin pointed out how the pollution of our environment represents such a case: while it is rational for an individual to indefinitely profit from activities that produce the pollution of the environment as byproducts, it is not rational for the collective as a whole to engage in such activities beyond a point where their aggregated effects produce a net disadvantage in the balance of benefits and costs [18].

On the one hand, many economists saw a solution to the problem of negative externalities, such as a polluting factory, in the workings of the invisible hand of the market. Basing their arguments on Coasian bargaining [37], economists argued that when an economic activity produces some externality, a market on the externalities, allowing for a bargain between the parties involved, will reach a Pareto efficient outcome. Nonetheless, as Hardin correctly assumed, for many cases of pollution of natural resources—such as our rivers, seas, and atmosphere—defining and enforcing clear property rights would seem difficult, if not impossible. Furthermore, as Coase himself pointed out, in most cases of polluting externalities, the spread of the impacts among a large number of individuals would make the organization of a bargain extremely costly, making transaction costs extremely high. Elinor Ostrom systematized these observations, pointing out the limits of privatization in solving commons dilemmas when: (1) resources are nonstationary, (2) resources are global or have a large geographical extension, (3) it is difficult to place boundaries and protect the private property, and (4) resource flow is unevenly distributed in both space and time [27]. Many common resources, such as oceans, water basins, coral reefs, animal habitats, the atmosphere, and many of Earth's ecosystems, are difficult, and indeed at times impossible, to privatize. As a result, the problem of negative externalities in many commons dilemmas seemed to be simply unsolvable via the simple mechanism of the market.

On the other hand, Hardin eventually became a supporter of the public management of commons, arguing that "if ruin is to be avoided in a crowded world, people must be responsive to a coercive force outside their individual psyches, a 'Leviathan' to use Hobbes' term" [18]. In this picture, the authority, as a Leviathan, must act in the collective interest by modifying the structure of incentives producing the externalities and restore optimal coordination in the management of the commons. For economists advocating a bigger role of the state in solving externalities, the action of government has to take the form of Pigovian taxation, designing incentives or establishing sanctions to change the structure of payoffs in the game and restore coordination between the actors involved, so as to internalize the externalities and prevent a less-than-efficient outcome from being realized. In this way, the actors can carry on their activities based on the exploitation of the common resource without depleting it. However, Ostrom claimed that turning commons into public goods was bound to face some shortcomings when (1) creating new institutions may turn out to be slow or difficult, (2) creating new institutions has high costs, and (3) institutions may demonstrate inefficient in managing the commons [27]. Interestingly, within environmental governance, the "market" solution, of Coasian inspiration, is at the base of contemporary cap-and-trade systems [37,38]. These markets work by setting a maximum threshold of emissions within a country and allowing companies to trade in emission permits according to their productive necessities. In contrast, the "Leviathan" solution, of Pigovian inspiration, grounds contemporary forms of carbon taxation [39]. In this case, the negative externalities are internalized through a different form of carbon

pricing: a Pareto efficient outcome is secured by setting a tax on emissions equal to the social costs generated through the polluting activities.

### 3.4. Conventional Solutions to the Problem of Many Hands: Organization and Authority

As we saw, within the field of applied ethics, van de Poel defines the Problem of Many Hands as a dilemmatic disconnect between individual and collective responsibility. Within his philosophical framework, the Problem of Many Hands is framed as resulting from a failure to effectively distribute responsibility in a group [20]. The argument develops by pointing out that whenever a collection of agents lacks a proper organizational structure, no single actor has a formally defined role with a respective array of task-responsibilities. This is a consequence of the impossibility of properly discharging the collective responsibility among an uncoordinated collection of agents, since the group lacks an organizational structure for effectively distributing responsibility at the individual level. The main proposal of van de Poel is then to suggest that, to prevent the occurrence of the Problem of Many Hands, a collective needs a better organizational structure for efficiently distributing responsibilities among the various actors. Accordingly, van de Poel seems to follow the work of Grossi, Royakkers, and Dignum in *Organizational Structure and Responsibility* by claiming that increased organization is to be achieved through the establishment of clearer *authority*, defining a hierarchical structure of responsibility delegation from a decisional center; better *coordination*, granting an increased flow of relevant information and knowledge between the actors involved; and increased *control*, securing a stricter supervisory activity [40,41]. According to van de Poel, we can sketch a taxonomy of three different types of groups to which the Problem of Many Hands applies in cases of prospective responsibility [20]:

> *1. Organized groups (also sometimes called 'corporate agents') that can formulate and adopt collective aims by a collective (decision) procedure;*
>
> *2. Collectives involved in a joint action. The joint action is characterized by a collective aim that is in some sense [ . . . ] shared by the members of the collective;*
>
> *3. Occasional collections of individuals that lack a collective aim but that nevertheless can be reasonably expected to form a collective in one of the two above senses to avoid harm or to do good.*

Van de Poel suggests that, as one moves from organizations down to collectives and collections, the progressive fragmentation of agency and the resulting impossibility to distribute responsibilities back at the individual level creates the conditions for the emergence of responsibility gaps like the Problem of Many Hands. Therefore, preventing responsibility gaps from occurring requires organizing a group in a hierarchical structure that is centered around authority, coordination, and control.

While these conditions constitute the basis for the design of a clearer organization within hierarchical entities, like corporations or public administrations, it should nonetheless be noticed how such conditions are ill-suited to provide a proper ground for the coordination of governance networks in the management of the environment. In fact, a centralized conception of vertical organization best applies to traditional public administration where hierarchical trees and command and control practices define the structure of task-delegation within a group of public officials [15]. However, crucially, governing social–ecological systems confronts administrations with complex problems that are difficult to solve by a unique decision-making center [7,10]. The analysis and the management of complex feedbacks between social and ecological systems require the aggregation of a multiplicity of actors from public, private, and civil society sectors that provide a diverse range of expertise in articulated knowledge domains. Accordingly, the complexity of social–ecological systems is increasingly mirrored by the complexity of governance networks. This creates a new set of challenges at the substantive, strategic, and institutional levels: different actors hold different perceptions of policy problems, they follow different interests involving different and sometimes contrasting strategies, and finally, decision-making spans across different institutional settings, often with the superimposition of many accountabil-

ity mechanisms [15]. Such interdependent structures clash against a vertical organization of decision-making. As Kljin and Koppenjan argue "mutual dependencies make it impossible for each of the involved actors to act in isolation, or as principals and agents" [15]. This structural interdependency renders it difficult to organize governance networks along hierarchical lines. Accordingly, the governance of social–ecological systems has taken an increasingly polycentric character, where multiple and diverse decision-making centers interact through a hybrid matrix of competitive and cooperative ties.

## 4. Results and Discussion

### 4.1. The Role of Cooperative Governance in Managing the Climate Commons

One of the biggest merits of Elinor Ostrom has been the redefinition of our understanding of commons situations. During her career, she helped to establish a third theoretical solution, between market and state proposals, to the Tragedy of the Commons.

#### 4.1.1. Polycentricity in Commons Situations

Where the conventional theory of collective action predicted that, when faced with a commons dilemma, the actors would inevitably run into the destruction of the shared resource if not regulated by an external institution, the work of Elinor Ostrom, starting with her essay *Governing the Commons*, focused on providing empirical and theoretical insights to show that this was not an inevitable outcome [21]. Indeed, on many occasions, actors faced with a commons were able to reach an agreement among themselves and mutually enforce a contract that efficiently allocated the resource among the participants. What Ostrom discovered was that the set of assumptions made by neoclassical economists, which framed the commons situation as a game played by self-interested actors striving to maximize immediate utility and not engaging in communication, did not apply in many real-world situations. Ostrom and her team showed that agents, within a repeated game and allowed to have face-to-face communication, were shown to be "extremely successful in increasing joint returns" [42]. By repeating the game, the communication between actors allowed for the *emergence of collective forms of learning and normativity*: the emergence of the reputation of players, the emergence of trust in other players, and the emergence of mutual monitoring and sanctioning behaviors. In this way, the actors were able to devise and enforce a cooperative strategy, allowing them to reach Pareto efficient allocation of resources [21]. This theoretical insight allowed Elinor Ostrom to elaborate with Vincent Ostrom, her husband and colleague at Indiana University, a theory of polycentric governance, where decentralized, multilevel, and cooperative decision-making grounded a new understanding of institutional networks [8,42,43]. The Ostroms framed polycentric systems as being "characterized by multiple governing authorities at differing scales rather than a monocentric unit" where each governance unit "exercises considerable independence to make norms and rules within a specific domain" [42]. Polycentric systems were originally conceived by Vincent Ostrom as redundant governance systems where the compresence of competition and cooperation among decision-making centers was able to secure levels of dynamism and coordination at the same time. What Elinor Ostrom contributed was a dynamic understanding of how increased cooperation can emerge in the face of commons dilemmas. As the analysis of commons dilemmas had already brought to the fore, the progressive establishment of cooperative networks within polycentric systems presents the double advantage of allowing mutual learning between actors and fostering the emergence of coordinated action by means of shared normative structures setting common goals and rules. In fact, as Ostrom claimed, polycentric systems constitute a governance architecture that is likely to "enhance innovation, learning, adaptation, trustworthiness, levels of cooperation of participants, and the achievement of more effective, equitable, and sustainable outcomes at multiple scales" [42]. In what is perhaps the most in-depth study of polycentric governance of climate change, Jordan et al. frame polycentricity as a theory built around five propositions [44]: "(1) Governance initiatives are likely to take off at a local level through processes of self-organization; (2) Constituent units are

likely to spontaneously develop collaborations with one another, producing more trusting interrelationships; (3) The willingness and capacity to experiment is likely to facilitate governance innovation and learning about what works; (4) Trust is likely to build up more quickly when units can self-organize, thus increasing collective ambitions; (5) Local initiatives are likely to work best when they are bound by a set of overarching rules that enshrine the goals to be achieved and/or allow conflicts to be resolved".

However, where polycentricity has shown great promise at small- and mesoscales, many have voiced caution regarding the possibility of governing a global phenomenon like climate change cooperatively [45,46]. In this regard, Felix Ekardt has argued that cooperative networks work best only when the "cooperation of other participants is to be expected, when the situation is manageable, and norm violations are noticed and sanctioned"; all of these characteristics are problematic to assume in the global governance of climate change [46]. Nonetheless, some considerations might contribute to weakening the concerns around the development of cooperative action in tackling the climate crisis. In fact, despite the predictions of classical game theory, we assisted in recent decades to the creation of a myriad of cooperative initiatives in climate governance, from public–private partnerships to transnational networks of municipalities and regions. The United Nations Environmental Program currently counts 269 international networks of non-state actors in its Climate Initiatives Platform. Accordingly, these numbers contribute to present some evidence that the existence of "conditional cooperators" in the climate commons is far more widespread than assumed by rational choice models. Therefore, faced with the rapidly growing reality of cooperative governance networks, the most pressing question seems to be no longer whether such governance architectures could play a role in the management of climate change, but which role should we assign to them.

### 4.1.2. Climate Action: The Complementary Role of Cooperative Governance Networks

In addressing the challenge of the environmental governance of climate change, Ostrom has argued that conventional approaches that strive for the creation of global institutions have so far turned out to be too slow for the urgency of climate action, global regulation without local participation is bound to be ineffective, and finally, universal norms are often unresponsive to contextual situations and problems [9]. Within the fight against climate change, creating global institutions for governing the sustainable transition has proved to be extremely difficult so far. Since the 1990s, transnational efforts to converge on a shared and legally binding agreement between world governments have largely failed. Starting from the Rio Conference in 1992, the collective effort to create a global institution that can enforce a shared body of rules in tackling climate change has fallen short. In particular, as the Kyoto Summit in 1997 failed to gather widespread political support around common measures and regulations against global warming, there has been increasing recognition that environmental governance can benefit from a more cooperative and horizontal structure. The limits of the universalist approach of the Kyoto Protocol have been at the base of the different approach toward environmental governance championed within the 2015 Paris Agreement. This new international agreement moved away from the top-down logic of treaties and shifted toward a more flexible and bottom-up model, based on Nationally Determined Contributions, where targets, plans, and mutual monitoring mechanisms have to be set in place in the absence of any higher-order institution. This more flexible mechanism has allowed for a much larger commitment, with 191 countries and the EU among its signatories. As this shift away from rigid governance structures can be traced back to a form of *realpolitik*, it is also the case that Ostrom's work has brought new awareness to the potential of cooperative governance when dealing with the climate crisis [44]. Nonetheless, Ostrom always warned against the tendency to believe in policy panaceas that advocated for a single solution to the management of social–ecological systems [47]. In fact, the theory of polycentric governance was never intended to be the only answer to the challenge of meaningful climate action. In an important sense, Ostrom's main critique of the standard top-down approach that advocated for the creation of a

global institution for tackling climate change is that such a theory is too one-sided and it disregards the evolutionary dynamics of cooperation. In this regard, a theory of bottom-up and polycentric governance should be considered a necessary complement to top-down and centralized approaches for three main reasons. First, where top-down theories tend to provide a static answer to the challenge of climate change, usually framed in the form of abstract institutional architectures with a universal reach, Ostrom's approach can bring forth an *evolutionary understanding of institutional emergence* that is based on increasing co-operative ties among a differentiated set of local actors that progressively strengthen their mutual trust, align their goals and values, and only ultimately come to a shared framework of norms and rules. In this sense, the Kyoto Protocol represented an attempt to put the cart before the horse by proposing a universal normative structure, without the previous establishment of a meaningful body of cooperative ties based on mutual trust, shared goals, and aligned values. In this respect, the genealogical development of the Sustainable Development Goals and the bottom-up structure of the Paris Agreement marked a step forward in the comprehension of the evolutionary character of institutional emergence. Ostrom's theory of institutional development can therefore provide a better understanding of the *process* through which we arrive at the creation of shared institutions [22]. Second, cooperative governance networks are essential for providing a *bottom-up structure of local participation*, which is essential to complement the top-down imposition of a set of global regulations. As Ostrom pointed out, the institutional costs of regulatory enforcement are bound to be unsustainable without the creation of collaborative networks for climate action at every governance scale [9]. Local participation, from neighborhood initiatives to transnational municipal networks, is key for complementing top-down regulations with bottom-up cooperative action. In this regard, the emergence of cooperative networks of climate action at every scale has contributed to disprove the classic assumption of rational choice theory, which predicts that no actor faced with a commons dilemma will change his behavior unless an external authority enforces rules from above [9]. Governance net-works, such as the Global Covenant of Mayors or the C40 Cities Climate Leadership Group, have proven effective at gathering widespread political support around climate initiatives. Furthermore, sub-state actors have often proven themselves capable of leading the way in setting ambitious targets of emissions reductions that far exceed those of national govern-ments [44,48]. Even if we currently lack clear data for measuring the effectiveness of such initiatives, the progressive construction of shared commitments, data sets, research and innovation programs, and financing platforms represents an encouraging first step in the elaboration of cooperative strategies for flexible climate adaptation and effective climate mitigation [49–51]. Third, where centralized institutions can create stable, predictable, and durable governance architectures, polycentric networks can supplement the relative rigidity of top-down organizations with *increased levels of institutional flexibility* [5,52]. The advantages of adopting such a polycentric structure rely on the increased adaptiveness, institutional flexibility, and resilience of governance networks. In this respect, polycen-tric networks present a larger potential for establishing a social–ecological fit between institutional architectures and ecological interlinkages within the Earth's system. The polycentric, redundant, and flexible nature of cooperative governance networks is better suited to responding more swiftly and adaptively to evolutionary changes in complex social–ecological systems [13]. As Oran Young has argued, "as we move deeper into a world of complex systems characterized by non-linear change, bifurcations and emergent properties, there is a growing premium on creating governance systems that are agile or nimble in responding to changes in the issue areas they address" [52]. Accordingly, a value-driven and goal-based model of climate change governance could grant political accountability in setting climate targets while allowing for a level of policy flexibility that can better address the local differentiation of social and ecological conditions in the Earth system. To be sure, polycentric governance, with its emphasis on diversity and multiplicity in governance theory, can lead to institutional disorder and uncertainty when left unchecked [53]. Accordingly, as Young emphasizes, the design of climate governance

architectures must rely on the pragmatic balance between the dynamic benefits of policy fragmentation and the stabilizing effects of policy hierarchization [54]. In this respect, policy systematization, prioritization, and integration are essential tools within the process of institutional emergence [55]. However, institutional simplicity by means of excessive hierarchization risks reducing the institutional fitness to govern the complex nature of social–ecological interlinkages within the Earth system. Accordingly, we should strive to maintain a balance between "the perils of institutional reductionism and institutional overload" [54]. It can be argued that two great pragmatist lessons lie at the heart of Elinor and Vincent Ostrom's theory of governance: the refusal of untenable dualisms balkanizing the theoretical space in supporters of states or markets, centralization or decentralization, and the proposal of a theory of governance based on a dynamic understanding of collective agency as a process of institutional emergence.

### 4.2. Framing Responsibility in the Cooperative Governance of Climate Change

As argued, a large scholarly literature has been accumulating on how cooperative governance offers a promising approach in the management of social–ecological systems in the face of climate change. Come to this point, some problems can be raised: if cooperative governance networks are not organized along hierarchical lines, how can collective responsibility be distributed back to individual actors in the absence of a central authority? Can governance networks properly discharge the collective responsibility for preventing climate change? How do these networks have to be designed in order to allow for the coordinated agency necessary to distribute responsibilities across a collective? This section will then take charge of laying the building blocks of such a theory of collective responsibility in governance networks by grounding it on the social ontology of shared agency [56]. Once this is accomplished, the ultimate goal will be to propose a theory of cooperative governance that can avoid the emergence of responsibility gaps like the Problem of Many Hands.

#### 4.2.1. The Shortcomings of the Hierarchical Model

Let us, first, recapitulate the terms of the problem: humanity is the leading cause of climate change; this fact constitutes a prospective responsibility, i.e., a responsibility toward the future, to prevent this environmental crisis from occurring. As previously argued, prospective responsibility obtains when a societal actor is capable of moral agency, is causally efficacious in preventing the outcome to occur, and bringing about the outcome is normatively wrong. Van de Poel argues that only a form of organization based on *authority*, *coordination*, and *control* can properly discharge its prospective responsibility by creating effective mechanisms for distributing responsibilities at the individual level. Once this conclusion has been established, most authors within environmental ethics have focused on the role of national institutions in mitigating climate change. In fact, within this *hierarchical* approach, only national governments are regarded as bearing the collective responsibility for preventing the climate crisis due to their ability to properly discharge this responsibility through an organized and effective structure of decision-making, and therefore, be causally efficacious in solving it [17,20,34,36]. Accordingly, individual persons—but also other societal actors, which can be said to have an organized agency like firms, municipalities, regional institutions, etc.—are believed to lack a full responsibility to address the climate crisis, as they cannot be regarded as effective at mitigating the effects of global warming. Therefore, the argument continues, national governments bear the full responsibility to establish a set of global measures to grant a sustainable transition. Unfortunately, this solution is not fully satisfactory. What these authors seem to underestimate is the fact that the problem of responsibility is just moved to a higher level, but its structure remains the same since up until this point governments were not able to converge on the creation of a global institution. If we follow this hierarchical model, the absence of a global institution that can distribute collective responsibility implies the implosion of the individual responsibility of national governments to prevent climate

change. Hence, it seems that governments are facing a paradigmatic case of the Tragedy of the Commons and, consequently, a paradigmatic case of the Problem of Many Hands. In fact, even if nation states could be, but ultimately are failing to be, causally efficacious in governing a sustainable transition (second condition for PR), it still seems problematic to regard such a failure as a form of wrong-doing (third condition for PR) because unilaterally restricting the consumption of the commons can be seen as both ineffective and unfairly competitively disadvantageous, and continuing to consume it as neither sufficient nor necessary to cause climate change. Accordingly, the international governance of climate change can be seen as another instance of collective responsibility without individual responsibility, and therefore, as an instance of the Problem of Many Hands. Reached this point, we encounter a dead-end: only national institutions can be causally efficacious in the transition and only to the degree that they converge on a global institution that distributes the collective responsibility for climate action among them; such an institution is missing, making the single countries ultimately not responsible. Which options remain available in this scenario? At this point, it is important to notice that a hierarchical approach rests on two basic assumptions:

- *Pragmatic assumption*: only national or international institutions are causally efficacious in tackling climate change;
- *Theoretical assumption*: only a hierarchical structure organized around a decision-making center can effectively distribute responsibility.

At the pragmatic level, it can be pointed out how between the first 100 global economic revenue collectors, only 29 are states, while 71 are corporations [57]. Even setting aside the mere question of economic power and resources, a study by the Climate Accountability Institute showed that just 20 companies have contributed to 35% of the global greenhouse gas emissions since 1965 [58]. Additionally, one can also consider sub-state institutions as a promising vector for effective change in sustainable governance; for instance, as Jordan argues, "more than 100 regional governments have committed themselves to reducing emissions by at least 80 per cent by 2050, a target exceeding that of most sovereign states" [44]. In fact, we assisted in recent decades to a flourishing of climate networks between actors as diverse as regions, such as the Governors' Climate and Forests Task Force; municipalities, such as the C40 Cities Climate Leadership Group, the Global Covenant of Mayors, and the International Council for Local Environmental Initiatives; and more broadly, a vast array of public–private partnerships. Once this is taken into consideration, it seems clear that a much larger range of social entities, from corporations to subnational actors such as regions and municipalities, can be causally efficacious in tackling climate change. Furthermore, at the theoretical level, the idea that only an organization structured along hierarchical lines can discharge our collective responsibility for climate mitigation is also questionable. Therefore, the main challenge of the next pages will be how to achieve an effective distribution of responsibility in cooperative governance networks. Our strategy will be to take the philosophy of shared agency developed by Michael Bratman in his Shared Agency: A Planning Theory of Acting Together [56] and argue that it can provide a theoretical grounding for the design of an organized distribution of moral labor in governance networks, so as to allow for the creation of responsible governance.

### 4.2.2. A Theory of Shared Agency: Five Design Principles for Cooperative Governance

Michael Bratman has spent his career working on a grand project aimed at the articulation of a full theory of human agency. Since his seminal work *Intention, Plans, and Practical Reason*, Bratman has focused on the crucial role of intentions in defining what constitutes the essential nature of our agency [59]. According to Bratman, an intention is essentially a plan to achieve a goal. Accordingly, what sets intentions apart from desires is their peculiar role in practical rationality to settle our conduct through time: intentional action does not derive from responding to the momentary whims of the will, but from following those ends that we decide to treat as the reasonable guides of our action through life. In the vocabulary of Bratman, intentions are characteristic psychological planning

states that constitute higher-order, conduct-controlling pro-attitudes that settle upon deliberation our cross-temporal agency on certain goals [59–61]. For Bratman, every time we act intentionally, we respond to a cognitive structure of norms of intentional rationality, such as norms of (1) *plan–belief consistency*, as plans should be consistently grounded on our beliefs; (2) *means–end coherence*, as plans should be coherently supported by subplans that devise the right means to our ends; (3) *plan agglomeration*, as plans should consistently add together in a coordinated structure of agency over time; and finally, (4) *cross-temporal stability*, as plans should be stable in order to organize agency through time.

After sketching this general picture of intentional action, it is then possible to proceed to frame cooperative agency as a form of shared intentionality. As a matter of fact, Bratman has made a major contribution to the field of social ontology by creating a theory of shared agency that is grounded on the role of intentions in coordinating cooperation between agents [56]. According to Bratman, collective action can be analyzed under the lens of *shared intentions*; sharing a goal with others, in this perspective, constitutes the basic glue of sociality. In its most simple description, when a group of agents takes on a collective action based on a shared aim, we can formalize the intention of each of the members as expressing "I intend that we J" (where J is the shared activity): this structure of practical rationality is what allows the embedding of individual actions in a collective endeavor, and thus, to have *intermeshing intentions*. Bratman's thesis is that, as the normative structure of individual intentions is rich enough to grant intrapersonal coordination of individual agency across time, the very same normative structure can allow interpersonal coordination of individual agency across the social space. This mirrors the Nagelian recognition that we are, as rational agents, under the necessity of coordinating ourselves both intra-personally across time and inter-personally across social interactions [62]. Therefore, the same norms of practical rationality described above can supply the normative structure of our cooperative agency [56]. In this way, for Bratman, the four norms of individual practical rationality give rise to four associated norms of social plan–belief consistency, social means–end coherence, social plan–agglomeration, and social cross-temporal stability (or *social consistency*, *social coherence*, *social agglomeration*, and *social stability*). Therefore, we come to a crucial question for the development of a theory of cooperation: which are the essential rational conditions for achieving a consistent, coherent, and stable shared agency? Bratman's answer is that our shared agency meets the criteria for social consistency, coherence, and stability when these five conditions apply: (1) *intention condition*: each intends that we J and the intentions of each are interlocking (each intends to J by way of the intention of each that we J) and reflexive (each intends that we J by way of their own intention that we J); (2) *belief condition*: each believes that if the intentions of each in favor of J are persistent and interdependent, we will be effective at J-ing; (3) *interdependence condition*: each continues to intend that we J only if each continues so to intend such that there is interdependence in persistence; (4) *common knowledge condition*: it is common knowledge that 1–3 is occurring; (5) *mutual responsiveness condition*: each adapts their relevant subplans and actions by way of public mutual responsiveness to each other's sub-plans and actions in a way that keeps track of the shared intention to J by means of our intermeshing plans. When a collective agency is organized around these five conditions, we reach a form of cooperative agency. Hence, it can be suggested that Bratman's theory of shared agency can provide a rational structure for sketching some *design principles for cooperative governance*. Indeed, within Bratman's theory of shared agency, cooperation is bound to lose its gluing power as the number of decision-making centers scales up, but this does not imply that cooperation is less effective as we scale up the dimension of the governance units over which decision-making centers preside. It can then be advanced that governance networks are cooperative structures insofar as:

1.　actors share a goal and elaborate interlocking and reflexive policies;
2.　actors believe that if the policies are persistent and interdependent, the network will be effective in reaching the goal;
3.　such policies are interdependent in persistence;

4. the network grants common knowledge to all actors by way of relevant information flow;
5. actors achieve mutual responsiveness in elaborating subplans, so as to achieve inter-meshing of plans.

These conditions represent a set of practical rationality norms for the coordination of agency within cooperative networks and, it can be argued, they provide a set of design principles for cooperative governance networks. To the extent that polycentric networks are structured in such a way, they can be said to act cooperatively. Once these conditions apply within a governance network, the group can engage in a shared deliberation about the distribution of responsibilities among its members. Such a shared deliberation is a form of shared agency, first, because it is embedded within the shared intentional activity, second, because such deliberation is itself a form of shared intentional activity, and finally, because the proposals made within a shared deliberation are raised from within a structure of shared commitments to a common goal [56]. Therefore, when a collective is faced with a prospective responsibility within a cooperative agency structure, Bratman's theory provides the actors with the rational instruments for engaging in a shared deliberation that provides an agreed-upon policy that distributes responsibilities among the participants [56]. The five design principles for cooperative governance networks represent functional criteria for avoiding the fragmentation of agency and, hence, they constitute essential requirements for preventing responsibility gaps like the Problems of Many Hands. We then take Bratman's theory to provide the rational foundation for a theory of cooperation in governance networks. The capacity to effectively discharge the collective responsibility for preventing climate change is thus met without reference to an authority that delegates tasks, but by a shared deliberation based on common goals; interlocking, persistent, and interdependent policies; common knowledge; mutual responsiveness; and therefore, intermeshing plans.

### 4.2.3. Responsibilization: A Processual Account of Moral Change

One important consequence of developing this analysis of responsibility within cooperative governance is that our prospective responsibility for climate action can no longer be considered dependent upon a higher institution that takes charge to distribute it. Hence, the theoretical assumption of centralized approaches, according to which only a vertical institution can effectively discharge responsibility, has ultimately been demonstrated to be unwarranted. The moral consequence is that, at this point, responsibility falls back into the hands of the many actors that can be causally efficacious in preventing climate change by cooperating. As it was previously claimed, there is no reason for holding corporations and subnational actors like regions and municipalities as not causally efficacious in tackling climate change. This recognition amounts to a redistribution of moral labor from governments alone to a much larger array of societal actors, which share with these the prospective responsibility to coordinate and cooperate in order to mitigate the effects of climate change. In this regard, cooperative networks will vary in their degree of normative alignment: from relatively fragmented and voluntary forms of loose cooperation based on shared goals to increasingly organic and binding forms of tight cooperation, involving the emergence of shared normative practices of value setting, value prioritization, and finally, value operationalization by means of the systematic organization of an institutional body of norms. In a pragmatist spirit, we should see collective responsibility not only as an abstract requirement of practical reason but also as a concrete instance of moral evolution, as an emergent and continuous process of responsibilization in the face of a new societal challenge. Just as Ostrom provides us with an economic theory of institutional emergence in the face of social dilemmas, pragmatism can be regarded as complementary to Ostrom's analysis in proposing an ethical theory of moral emergence in the face of new practical problems. For this reason, we should avoid framing responsibility exclusively as the act of responding to abstract and universal reasons of morality; instead, we should complement it with an understanding of responsibility as a societal process of responsibilization in the face of the emergent threat of climate change. The concern for the top-down establishment of a series of moral and legal norms should therefore be accompanied by the articulation of

a bottom-up process of decision-making that is characterized by participatory, transparent, and flexible procedures that allow for the development of shared goals, values, and norms.

To conclude, we should redistribute the moral responsibility for swift climate action from national governments to a much larger array of actors encompassing firms, municipalities, and subnational regions. This responsibility is based on their potential to be causally efficacious in preventing climate change and in their ability to create a spectrum of cooperative structures that can properly discharge the collective responsibility for climate action through shared policies. Therefore, states, regions, cities, and firms are not discharged of their individual responsibility to act until the establishment of a global institution. Accordingly, this implies a great reduction in the severeness of the moral dilemma that is the Problem of Many Hands regarding climate change. Where interests, goals, or values are aligned, the creation of cooperative networks should be regarded as a promising way to organize a process of responsibilization within the global governance of climate change. Waiting for a global agreement to discharge our responsibility to act might be a strategical failure and indeed a morally unwarranted conclusion. Therefore, the prospect of meaningful climate action at the global level is considerably expanded, even if it can still be difficult to attribute such a prospective responsibility to individual persons. Nonetheless, as many authors have emphasized, individuals still retain a prospective responsibility as citizens to mobilize in order to pressure states, regions, and cities to take serious measures to tackle the moral and ecological crisis that is climate change [20,25,63]. Furthermore, even if individual persons cannot be said to bear the full responsibility for climate action, it might as well be a question of moral integrity to be consequential with our political responsibilities and apply the sustainable behavior we ask of our governments to our individual lives [64,65]. Furthermore, it is possible to argue that individuals have a prospective responsibility as consumers to boycott, when possible, those corporations that are among the main contributors to climate change. These recognitions amount to a further weakening of the Problem of Many Hands regarding climate change, as the gap between collective and individual responsibility for single citizens is, ultimately, a matter of degree and not of sharp opposition. Furthermore, the fact that we might not be fully responsible for meaningful climate action at the individual level does not exclude the fact that we might find alternative ways of living sustainably that are still preferable and more meaningful. Indeed, the appreciation for nature, simplicity, and the ecological character of human life, while not part of what is morally required, acquires perhaps even more meaning in its gratuitousness.

## 5. Conclusions

In this article, I aimed to reconstruct how climate change has been formalized as a Tragedy of the Commons in economic theory and as a Problem of Many Hands in ethical theory. I then proposed a conceptual connection between these two dilemmas and claimed that whenever a rational failure like the Tragedy of the Commons occurs, a parallel moral failure occurs, namely, the Problem of Many Hands, since "free-riding" the commons is not irrational or irresponsible at the individual level, while it constitutes a rational and moral failure at the collective level. I then proceeded to analyze how classical solutions to both dilemmas, which are usually framed in terms of the establishment of vertical structures of decision-making, are not the only possible answer to the challenge of the responsible governance of climate commons. I take Elinor Ostrom's theory of polycentric governance as a promising candidate to complement this classical top-down model with a bottom-up approach based on horizontal structures of governing with increasing levels of cooperation. At this point, three questions have emerged: how can collective responsibility be distributed back to individual actors in the absence of a central authority? Can governance networks properly discharge the collective responsibility for preventing climate change? How must these networks be designed in order to allow for the coordinated agency necessary to distribute responsibilities across a collective? The theory of shared agency of Michael Bratman has provided, in this regard, the theoretical basis for sketching

five design principles for cooperative governance networks. I argued that such networks can properly discharge responsibilities by engaging in a shared deliberation when cooperative networks are built around a shared goal; interlocking, persistent, and interdependent policies; common knowledge; mutual responsiveness; and thus, intermeshing of plans. I further claimed that we should frame collective responsibility not only as an abstract requirement of practical reason but also as a concrete evolutionary process of responsibilization. In the face of climate change, cooperative networks will certainly evolve in their degree of normative alignment: from fragmented and voluntary forms of loose cooperation based on shared goals to increasingly organic and binding forms of tight cooperation that involve the emergence of shared normative practices of value setting, value prioritization, and finally, value operationalization by means of the systematic organization of an institutional body of shared norms. The article has then contributed to show how institutional emergence and moral emergence can be analyzed as two aspects of a process of collective responsibilization.

Faced with the limits of the "technical resources" offered by economic rationality in a Tragedy of the Commons, Hardin wrote that the world "requires a fundamental extension of our morality" [18]. For Hardin, this was to be found in the coercive power of a Leviathan; I hope to have shown a way in which our morality can be fundamentally extended within a cooperative structure of collective agency.

**Funding:** This research received no external funding.

**Institutional Review Board Statement:** Not applicable.

**Informed Consent Statement:** Not applicable.

**Data Availability Statement:** Not applicable.

**Conflicts of Interest:** The author declares no conflict of interest.

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
