# Peer review of "Collective Responsibility in the Cooperative Governance of Climate Change"

_sustainability, doi:10.3390/su13084363_

Round 1
Reviewer 1 Report
This is a well-written paper on an important topic: how to solve the problem of cooperative governance of climate change? The author claims that “climate change is addressable only insofar as we transition from a morality based on individual responsibility to a new conception of morality based on our co-responsibility for preventing the climate crisis.” For this purpose, the author refers to the theory of Michael Bratman.
Since Bratman did not have a large group of actors in mind, it is interesting to see how Bratman’s theory is combined with Ostrom’s idea of governance based on polycentric, multi-level and cooperative decision-making.
My main criticism of the paper is that it sidesteps the central problem of cooperative governance of climate change, although briefly mentioned in the conclusion (lines 616-617): “Certainly, this analysis leaves open the moral question of what constitutes a fair distribution of responsibilities”. One of the main reasons why the world is incapable to govern climate change along hierarchical lines is that there is disagreement over sharing the costs of climate policy. There is no reason to assume that the author’s idea of responsible cooperative governance along Bratman’s lines will fare better. If there is no shared agreement about what constitutes a fair distribution of responsibilities, then the actors do not really share a common goal.
To give the “bottom-up model, based on nationally determined contributions” (lines 402-403) as a successful example of an alternative to the “top-down logic of treaties” is wishful thinking. It exactly proves the need of binding treaties, as shown from the inadequacy of the NDCs. See Rogelj, J., M. den Elzen, N. Höhne, T. Fransen, H. Fekete, H. Winkler, R. Schaeffer, F. Sha, K. Riahi, M. Meinshausen Paris Agreement climate proposals need a boost to keep warming well below 2 °C, Nature, 534 (2016), pp. 631-639. UNEP (2020). The Emissions Gap Report 2020. United Nations Environment Programme (UNEP), Nairobi.
I therefore would like to see a discussion of the Montreal Protocol which has proven that old-fashioned hierarchical governance does work. See Barrett, S. (2008). Climate treaties and the imperative of enforcement. Oxford Review of Economic Policy, 24(2), 239-258.
In short, I would like to see more argumentation to convince me that the bottom-up approach is a better alternative to hierarchical approach.
Moreover, it is not entirely clear to me to what extent this is a descriptive or normative paper. Bratman describes how collective intentions and coordinated activities can exist, but does not prescribe any morality. In his work, Bratman does not discuss or argue for collective responsibility. So it remains unclear how we ought to act.
Some other points:
Page 10: 502-518 goes much too fast. No convincing arguments are given for the ‘impossibility of organizing governance networks along hierarchical lines’. Apart from political unwillingness I do not see the fundamental impossibility.
Line 61: the shortcomings of centralized regulation. Which?
Line 66-67: effective multi-level coordination across government scales. I would like to see proof or examples.
Line 218-220: In the discussion of Johnson: there are good moral reasons to act as an individual in a collective action problem. See Hourdequin, M. 2010. 'Climate, collective action, and individual ethical obligations', Environmental Values 19: 443-464. Marion Hourdequin, “Climate Change and Individual Responsibility: A Reply to Johnson,” Environmental Values 20, no. 2 (2011): 162. Hedberg, T. (2018). Climate change, moral integrity, and obligations to reduce individual greenhouse gas emissions. Ethics, Policy & Environment, 21(1), 64-80.
Line 231: Do ‘collective reasons’ exist?
Line 239: I know that it is a quote, but does collective responsibility exist?
Line 376-384: Ostrom recommended that collective action problems are addressed at the level where they occur. This would lead to different scales of governance just as environmental problems occur at different scales. I do not read Ostrom as that climate change can be solved by a layered system of governance.
Author Response
Dear Reviewer,
as a first remark, I would like to say that I appreciate the attention with which the article has been reviewed.
I will first address the main criticism, regarding the fact that the paper does not take position in the ethical debate on what constitutes a just distribution of burdens with regard to climate action. My answer is that the paper aims at responding to a different challenge pertaining the meta-ethical debate: who should we regard as responsible for climate action? My goal is to point out that the view according to which only national institutions bear the responsiblity to mitigate climate change is unwarranted. I then argue for a redistribution of moral labor to include regions, cities and firms among the actors that bear a responsibility to act. Given the urgency of tackling climate change, my aim is to argue that the absence of a global institution, which is universally agreed upon, does not release societal actors, at every scale, to take the responsibility to cooperate with whoever has common interests, goals and values to take immediate action.
Regarding the idea that bottom-up governance has a better chance of solving climate change, I do not commit to this view in the text. The aim of the paper is to argue for the necessity of bottom-up governance as a complementary model to the top-down logic of treaties. Regarding the fact that the Paris Agreement has been more successful than the Kyoto Protocol in gathering political support, I take this as an indication of what constitutes a better strategy in understanding the processual character of responsibilization at the global level. To me the idea of converging on a legally binding set of common regulations without previously reaching a degree of convergenge on goals, values, and respective priorities seems wishful thinking. In this respect the idea of a transition phase where we "govern through goals" seems crucial. Furthermore, the Paris Agreement has a mechanism for periodic global stocktake where national commitments are revised; there is therefore hope for averting a failure.
I am sorry for not discussing the Montreal Protocol in the paper. I felt there wasn't an organic space for its discussion in the text. I could perhaps do it in a note. My idea, shared with others, is that phasing out the use of CFCs was far easier than the complete transformation of the economy which climate change requires. Where Montreal was limited to a few industries, tackling climate change involves widespread industry change, to a degree that involves a rethinking of the entire model of development of our society. I doubt that such a decision can be taken top-down, without gathering a widespread support from the bottom-up.
In the paper I expanded greatly the discussion on the importance of bottom-up forms of cooperative governance.
Line 372-392: here I give an account of why it is difficult to organize governance networks vertically, especially when they are characterized by public-private partnerships.
Line 764-766: I thank you for indicating the papers of Hourdequin. I appreciated them greatly and they appear in the final part of the paper. Both authors seem to agree, in end, that the difference in responsibility between individuals and governments it's a matter of degree, not of sharp opposition.
Do collective reasons even exist? To the degree that we talk of what is "collectively rational" in a tragedy of the commons I would say they do. Just as in meta-ethics we talk of "impartial reasons", it can make sense to talk of collective reasons within the fiels of economics. Tom Nagel has written a great deal on the "external view" in his The View from Nowhere.
The debate on collective responsibilty is indeed a huge part of the meta-ethical literature in the last 20 years. For a good introduction, the Stanford Encyclopedia page on Collective Responsibility is, as often, very well done.
I thank you for your comments. I hope I managed to do them justice and they certainly contributed to make the work more solid. I remain at your disposal for any doubts or clarifications.
Best regards.
Reviewer 2 Report
The paper is interesting as it attempts to provide a structure for thinking about climate change action at a global scale. However, it is very difficult to follow as the text is circuitous and needs a great deal of tightening up. Frankly, I just got lost many times in the logic. Also- something to consider. Hardins thesis on the tragedy of the commons has been disputed for a long time- seems that it would be important to bring those critiques up in this document. This paper- could be interesting, but it needs a lot of work. And could maybe even benefit from action research that puts forward the ideas of elevating the voices of marginalized groups as well.

Author Response
Dear Reviewer,
I thank you for your comments on the paper. In order to make the structure of the paper more clear I added a section on Methodology, so to explicit better the conceptual structure of the paper. I hope this gives the reader a more clear sense of the development of the argumentation. I also wish to emphasize that in the paper I do not straightforwardly endorse Hardin's formalization of commons dilemmas. Indeed, in the text I side with Elinor Ostrom in criticizing the rational choice model of individual rationality informing Hardin's reading of commons dilemmas. I hope this extensive rework of the original material made the article more solid in content and clear in style.
Best regards.
Round 2
Reviewer 1 Report
Dear editor,
I believe the author and I will continue to disagree on the fundamental bottle neck for successful mitigation of climate change. I consider the article well-written and interesting, however.
Author Response
Dear Reviewer,
I made some minor changes to some sentences in the text in order to make it more clear in style and solid in argumentation. I thank you for your inputs, which have been certainly useful. While we continue to disagree, maybe we practiced what Richard Sennett calls a "differentiating exchange".
Best regards.
Reviewer 2 Report
Greatly improved and easier to follow. There are a few cases where sentences are long and could be shortened for clarity. A final read through by the author for clarity should be conducted before final submission.
Author Response
Dear Reviewer,
I did a final read through of the paper and I corrected some sentences here and there to make it more clear and readable. I thank you for your comments and attention.
Best regards.